# Targeting of NLRP3 inflammasome with gene editing for the amelioration of inflammatory diseases

Congfei Xu[1,2,3], Zidong Lu[1,2], Yingli Luo[4], Yang Liu[4], Zhiting Cao[4], Song Shen[1,2,3], Hongjun Li[1,2,3], Jing Liu[4], Kaige Chen[4], Zhiyao Chen[4], Xianzhu Yang[1,2,3], Zhen Gu [5] & Jun Wang [1,2,3,6]

The NLRP3 inflammasome is a well-studied target for the treatment of multiple inflammatory diseases, but how to promote the current therapeutics remains a large challenge. CRISPR/Cas9, as a gene editing tool, allows for direct ablation of NLRP3 at the genomic level. In this study, we screen an optimized cationic lipid-assisted nanoparticle (CLAN) to deliver Cas9 mRNA (mCas9) and guide RNA (gRNA) into macrophages. By using CLAN encapsulating mCas9 and gRNA-targeting NLRP3 (gNLRP3) (CLAN$_{mCas9/gNLRP3}$), we disrupt NLRP3 of macrophages, inhibiting the activation of the NLRP3 inflammasome in response to diverse stimuli. After intravenous injection, CLAN$_{mCas9/gNLRP3}$ mitigates acute inflammation of LPS-induced septic shock and monosodium urate crystal (MSU)-induced peritonitis. In addition, CLAN$_{mCas9/gNLRP3}$ treatment improves insulin sensitivity and reduces adipose inflammation of high-fat-diet (HFD)-induced type 2 diabetes (T2D). Thus, our study provides a promising strategy for treating NLRP3-dependent inflammatory diseases and provides a carrier for delivering CRISPR/Cas9 into macrophages.

[1] Guangzhou First People's Hospital, School of Medicine, South China University of Technology, 510006 Guangzhou, Guangdong, China. [2] National Engineering Research Center for Tissue Restoration and Reconstruction, South China University of Technology, 510006 Guangzhou, Guangdong, China. [3] Key Laboratory of Biomedical Materials and Engineering of the Ministry of Education, School of Biomedical Science and Engineering, South China University of Technology, Guangzhou International Campus, 510006 Guangzhou, Guangdong, China. [4] School of Life Sciences, University of Science and Technology of China, 230027 Hefei, Anhui, China. [5] Department of Bioengineering, California Nanosystems Institute, University of California, Los Angeles, CA 90095, USA. [6] Key Laboratory of Biomedical Engineering of Guangdong Province, South China University of Technology, 510006 Guangzhou, Guangdong, China. These authors contributed equally: Congfei Xu, Zidong Lu. Correspondence and requests for materials should be addressed to J.W. (email: mcjwang@scut.edu.cn)

The NLR family protein NLRP3 is a cytosolic sensor of exogenous pathogens and endogenous damage-associated molecular patterns (DAMPs)[1,2]. Upon activation, NLRP3 assembles with the adapter protein ASC and cysteine protease caspase-1 to form the NLRP3 inflammasome, resulting in the cleavage and activation of caspase-1. Activated capase-1 cleaves the precursors of IL-1β and IL-18 into mature forms and cause the release of several proinflammatory cytokines, including IL-1β and IL-18[3,4]. It was reported that the NLRP3 inflammasome plays critical roles in the initiation and progression of diverse inflammatory diseases[5–7]. Inhibition of the NLRP3 inflammasome signal has been shown to be effective in attenuating septic shock[8,9], peritonitis[8,10,11], Alzheimer's disease[12], atherosclerosis[13], T2D[14–16], multiple sclerosis[17,18], and gout[19], among other diseases. Thus, the NLRP3 inflammasome is an excellent target for the treatment of multiple inflammatory diseases.

Current strategies targeting the NLRP3 inflammasome mainly includes anti-inflammatory biologics targeting IL-1 signal (such as IL-1 receptor antagonist Kineret, neutralizing IL-1β antibody ILARIS and soluble decoy IL-1 receptor Arcalyst)[20–22], small-molecule inhibitors (such as glyburide and MCC950) and natural inhibitors of the NLRP3 inflammasome (such as omega-3 fatty acids, dopamine, and β-hydroxybutyrate)[8,11,16,18,23]. Biologics targeting the IL-1 signal only block the partial downstream signal of the NLRP3 inflammasome, and the compensation effect between different inflammatory mediators limits their therapeutic efficiency[20,24,25]. Except for the recently reported inhibitor CY-09 by Zhou et al.[26] most well-studied inhibitors usually inhibit the upstream signal of the NLRP3 inflammasome, not NLRP3 itself[26], which limits their efficiency and may causes unexpected side effects. CRISPR/Cas9, as a third-generation genome editing tool, can specifically and effectively disrupt or repair disease-causing genes by a single gRNA-directed Cas9 nuclease[16,27–29]. Using CRISPR/Cas9 to directly disrupt the key molecule-NLRP3 at the genomic level can not only completely inhibit the activation of NLRP3 inflammasome, but also avoid the potential risks of inhibiting off-target pathways of anti-inflammatory biologics and inhibitors. Development of a strategy to knock out NLRP3 with CRISPR/Cas9 is expected to be a more effective therapy for diverse inflammatory diseases[30].

Thus far, CRISPR/Cas9-modified T-cells in vitro have entered a clinical trial for the treatment of metastatic non-small cell lung cancer[31]. Adenovirus-associated or adeno-associated virus (AAV)-mediated delivery of CRISPR/Cas9 has been used to treat diseases such as hypercholesterolemia and Duchenne muscular dystrophy[32–35]. Non-viral delivery systems have also been developed to deliver the CRSPR/Cas9 plasmid, mRNA/gRNA or the Cas9/gRNA ribonucleoprotein complex for in vivo genome editing[36–48]. For example, Anderson et al. successfully treated FAH gene mutation induced-hereditary tyrosinemia using lipid nanoparticles and AAV-delivered mCas9/gRNA[37]. Wei et al. constructed a "core–shell" artificial virus (RRPHC) to deliver CRISPR/Cas9 plasmid for MTH1 gene disruption, significantly inhibiting SKOV3 xenograft tumor growth[36]. Thus, direct knockout of NLRP3 by CRISPR/Cas9-mediated gene editing is feasible for inflammatory disease treatment.

In this study, we report a systemic delivery system of CRISPR/Cas9 by encapsulating mCas9 and gNLRP3 into CLAN. CLAN is a type of PEG-b-PLGA-based nanoparticle assisted by cationic lipid BHEM-Chol for nucleic acid therapeutics delivery. In our previous work, we have delivered small interfering RNA, RNA aptamers and hepatitis B virus CpG into tumor cells, cardiomyocytes, macrophages or plasmacytoid dendritic cells with CLAN[10,49–51]. However, mCas9/gNLRP3 are different from other nucleic acid therapeutics, and the properties of nanoparticles impact the efficiency of drug delivery. Therefore, we create a library of CLANs of different surface charge and PEG density by adjusting the weight of the cationic lipid BHEM-Chol and mass fraction of $PEG_{5K}$-b-$PLGA_{11K}$ in polymers. We screen CLANs both in vitro and in vivo and select a preferable CLAN to deliver mCas9/gNLRP3 into macrophages, which ameliorates LPS-induced septic shock, MSU-induced peritonitis and HFD-induced T2D by disrupting NLRP3 in macrophages. Our study provides a promising strategy for the delivery of CRISPR/Cas9 into macrophages and treatment of multiple inflammatory diseases.

## Results

**Screening of a preferable CLAN for mCas9/gRNA delivery.** It was reported that the in vivo fates of diverse nanoparticles, including polymeric nanoparticles, are closely related to their nano-properties[52,53]. Rational design of the surface charge and PEG density of polymeric nanoparticles can enhance their internalization of target cells, including macrophages[53–55]. To fabricate a CLAN that is suitable for delivering mCas9/gRNA into macrophages, we designed a screening process (Fig. 1a).

First, we created a library containing 16 CLANs (CLAN11 to CLAN44) of different surface charge and PEG density. The components for CLAN preparation are shown in Supplementary Table 1. Briefly, the surface charge of CLAN was regulated by adjusting the weight of BHEM-Chol and the surface PEG density of CLAN was regulated by adjusting the mass fraction of $PEG_{5K}$-b-$PLGA_{11K}$ in polymers. Next, we characterized the properties of CLANs. The surface charge of CLANs ranged from 5.0 to 31.1 mV and was increased with the increase in BHEM-Chol (Supplementary Fig. 1a). To verify the PEG density of CLANs, we used ${}^1H$ NMR to detect the mass fraction of $PEG_{5K}$-b-$PLGA_{11K}$, which is PEGylated, and used it to represent the relative PEG density of each CLAN. As shown in Supplementary Fig. 1b, the mass fraction of $PEG_{5K}$-b-$PLGA_{11K}$ in polymers was decreased along with the increase in $PLGA_{11K}$. The diameter of each CLAN was similar (approximately 130 nm, Supplementary Fig. 1c). To facilitate the screening of CLANs with better macrophage uptake in vivo, we encapsulated Cy5-labeled siRNA (Cy5-siRNA) into CLANs to label them, and denoted them as $CLAN_{Cy5-siRNA}$. The encapsulation efficacy of Cy5-siRNA in CLANs was measured by high-performance liquid chromatography (HPLC) and was higher than 90% (Supplementary Fig. 1d).

Subsequently, we detected the macrophage uptake of $CLAN_{Cy5-siRNA}$. Macrophages of mice intravenously injected with $CLAN_{Cy5-siRNA}$ were isolated and subjected to fluorescence-activated cell sorting (FACS) analysis. The relative uptake quantity of $CLAN_{Cy5-siRNA}$ was calculated as described in the Methods section. As shown in Fig. 1b and Supplementary Fig. 2, CLAN42 showed the highest relative uptake quantity of macrophages from the peritoneal cavity, liver, peripheral blood, spleen and adipose tissue. This result was consistent with other reports that increasing the surface charge of nanoparticles can increase cellular uptake[53,54], but the surface PEG density of CLANs was not always the lower the better.

Next, we selected CLAN42 for mCas9/gRNA delivery. Four other CLANs (CLAN11, CLAN14, CLAN41, and CLAN44) of distinct surface charge and PEG density were selected as controls. The properties of CLANs encapsulating mCas9/gRNA were characterized (Supplementary Fig. 3). The surface charges of CLAN42 and two other CLANs (CLAN41 and CLAN44) were similar (31.5 mV ± 5.84 mV, 30.7 mV ± 5.61 mV, and 31.4 mV ± 6.14 mV, respectively), and higher than those of CLAN11 and CLAN14 (9.62 mV ± 4.11 mV and 10.8 mV ± 4.37 mV, respectively) (Supplementary Fig. 3a). Additionally, five CLANs had a

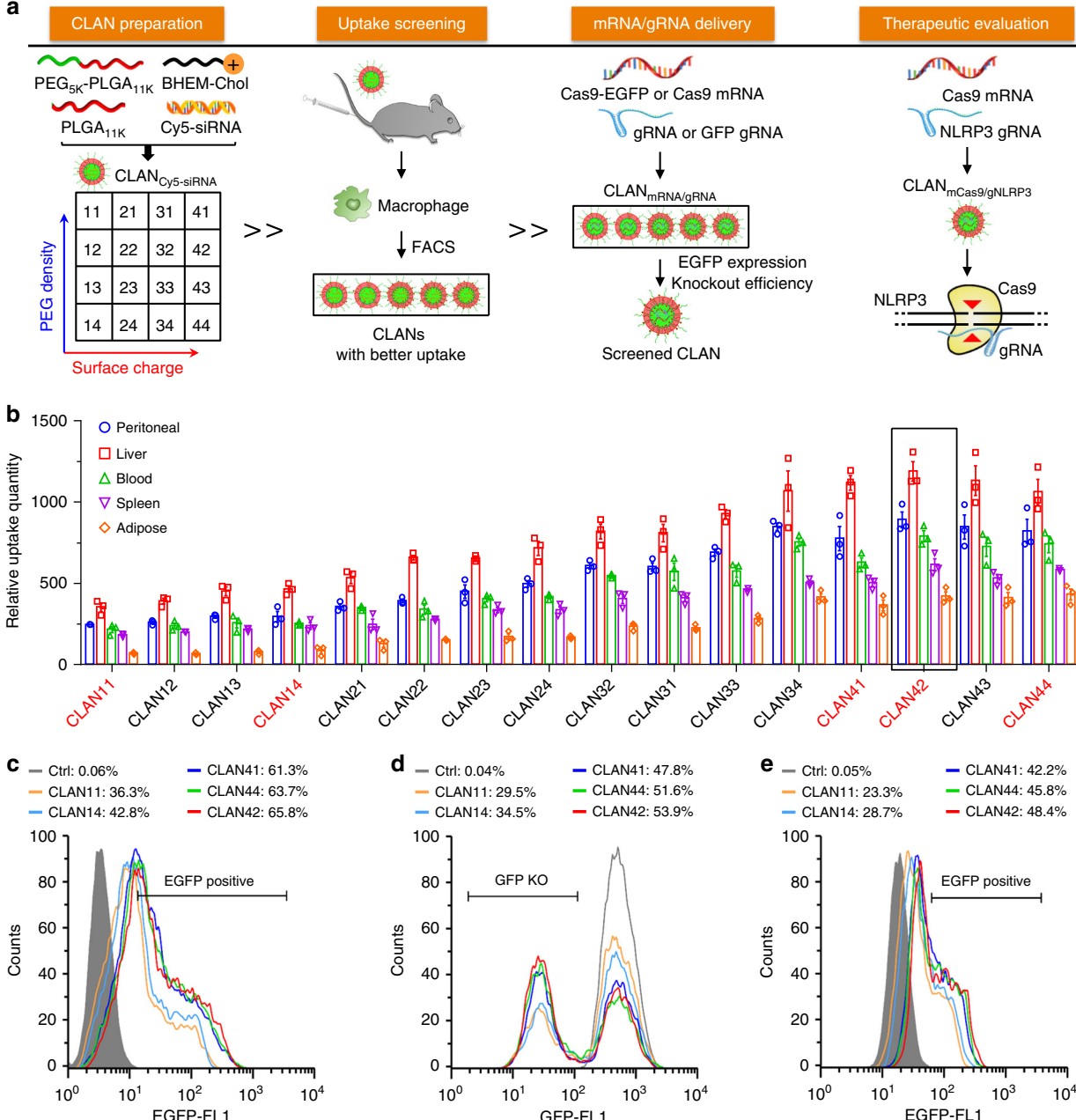

**Fig. 1** Screening of CLAN for mCas9/gRNA delivery to macrophages. **a** General scheme of this study. We first prepared a CLAN library of different surface charge or PEG density by adjusting the amount of BHEM-Chol or polymers (PEG$_{5K}$-$b$-PLGA$_{11K}$ and PLGA$_{11K}$). Next, we evaluated macrophage uptake of CLAN$_{Cy5-siRNA}$ by FACS. Screened CLANs were used to encapsulate Cas9-EGFP or Cas9 mRNA and gRNA. EGFP expression of CLAN$_{mCas9-EGFP/gNC}$ and the GFP-knockout efficiency of CLAN$_{mCas9/gGFP}$ were detected to determine which CLAN was better for macrophage gene editing. Finally, we encapsulated mCas9/gNLRP3 into screened CLAN to prepare CLAN$_{mCas9/gNLRP3}$ and used it for inflammatory disease treatment. **b** FACS analysis of the relative uptake quantity of each CLAN$_{Cy5-siRNA}$ in macrophages of mice injected with different CLAN$_{Cy5-siRNA}$. **c** FACS analysis of EGFP expression in BMDMs transfected with each CLAN$_{mCas9-EGFP/gNC}$. **d** FACS analysis of the GFP knockout efficiency of Raw264.7-GFP cells transfected with each CLAN$_{mCas9/gGFP}$. **e** FACS analysis of EGFP expression in peritoneal macrophages of mice injected with each CLAN$_{mCas9-EGFP/gNC}$. The data are shown as the means ± SEM of $n = 3$ (**b**) or are representative of three independent experiments (**c–e**)

similar diameter, morphology, and colloidal stability (Supplementary Fig. 3b-d). Five CLANs all could effectively encapsulate mCas9/gRNA, and the encapsulation efficiency was higher than 90% (Supplementary Table 2). We also tested whether CLANs could protect mCas9/gRNA in the physiological environment by incubating CLAN$_{mCas9/gRNA}$ with 20 µg ml$^{-1}$ RNase A. As demonstrated in Supplementary Fig. 4, mCas9 and gRNA both maintained their integrity with the encapsulation of five CLANs.

To test whether CLAN42 could effectively deliver mCas9/gRNA, we encapsulated Cas9 and enhanced green fluorescent protein (EGFP) co-expressing mRNA (Cas9-EGFP mRNA, or mCas9-EGFP) and negative control gRNA (gNC) into selected CLANs (CLAN$_{mCas9-EGFP/gNC}$). Bone marrow-derived macrophages (BMDMs) were transfected with different CLAN$_{mCas9-EGFP/gNC}$. As displayed in Fig. 1c, the CLAN42 transfection group showed the highest percentage of EGFP-positive BMDMs

(65.8%). Next, we detected the gene knockout efficiency by transfecting Raw264.7 cells (a macrophage cell line) stably expressing GFP (Raw264.7-GFP) with CLANs encapsulating mCas9 and gRNA-targeting GFP (gGFP) (CLAN$_{mCas9/gGFP}$). The percentage of GFP-knockout (KO) Raw264.7-GFP cells in the CLAN42 transfection group was the highest and reached 53.9% (Fig. 1d). We further confirmed the in vivo mCas9/gRNA delivery efficiency of CLAN42 by injecting mice with different CLAN$_{mCas9-EGFP/gNC}$. Similar to Fig. 1c, the percentage of EGFP-positive peritoneal macrophages in the CLAN42 injection group was the highest (48.4%) (Fig. 1e). Taken together, CLAN42 was the most effective CLAN in mCas9/gRNA delivery due to its highest ability of macrophage uptake, and CLAN42 was preferred to encapsulate mCas9/gNLRP3 (denoted as CLAN$_{mCas9/gNLRP3}$) for multiple inflammatory disease treatments.

**Localization and integrity of CLAN42 after internalization.** Before applying CLAN$_{mCas9/gNLRP3}$ for inflammatory disease treatments, we firstly studied its intracellular localization. We encapsulated Cy5-labeled mCas9 and gRNA into the screened CLAN42, denoted as CLAN$_{Cy5-mCas9/gRNA}$. Subsequently, BMDMs were transfected with CLAN$_{Cy5-mCas9/gRNA}$. As displayed in Supplementary Fig. 5, most of CLAN$_{Cy5-mCas9/gRNA}$ co-localized with EEA1-positive endosome at 0 h after transfection. This result indicated that CLAN$_{Cy5-mCas9/gRNA}$ was internalized by macrophages by the endocytosis pathway and accumulated in the endosome after uptake. With the elongation of incubating time from 0 to 24 h, the red fluorescence of Cy5 gradually separated with the green fluorescence of the endosome, indicating that CLAN$_{Cy5-mCas9/gRNA}$ escaped from the endosome into the cytoplasm. In addition, the number of the endosome at 24 h was significantly lower than that at 12 or 6 h. We speculated that the escape of CLAN$_{Cy5-mCas9/gRNA}$ from the endosome was due to the disruption of the endosome. Besides, Cy5 signal was not observed in the nucleus (DAPI positive), indicating that neither CLAN$_{Cy5-mCas9/gRNA}$ nor Cy5-mCas9 passed into the nucleus. The fluorescence of Cas9 protein was observed in the cytoplasm and the nucleus at 6 h after transfection and Cas9 protein in the nucleus was increased at 12 and 24 h, suggesting that Cas9 was translated from mRNA into protein in the cytoplasm, and subsequently entered the nucleus under the guide of nuclear localization signal (NLS) co-expressed with Cas9.

Subsequently, we analyzed the integrity of CLAN42 after internalization by macrophages with a Förster Resonance Energy Transfer (FRET) experiment, and selected FITC (maximal excitation: ~490 nm, maximal emission: ~525 nm) and Rhodamine (maximal excitation: ~550 nm, maximal emission: ~590 nm) as the donor and acceptor molecules of FRET couple, respectively. FITC-conjugated PLGA$_{11K}$ and Rhodamine-conjugated PLGA$_{11K}$ were used to fabricated CLAN42, denoted as CLAN$_{FRET}$. CLAN fabricated with only FITC-conjugated PLGA$_{11K}$ was denoted as CLAN$_{FITC}$ and CLAN fabricated with only Rhodamine-conjugated PLGA$_{11K}$ was denoted as CLAN$_{Rho}$. As shown in Supplementary Fig. 6a, at 521 nm, the fluorescence intensity of CLAN$_{FRET}$ was 6384, which was significantly lower than that of CLAN$_{FITC}$ (17713). At 592 nm, the fluorescence intensity of CLAN$_{FRET}$ was 18352, which was significantly higher than that of CLAN$_{Rho}$ (3250). These results indicated that the energy of FITC was transferred to Rhodamine when CLAN$_{FRET}$ was intact. Then, we transfected BMDMs with CLAN$_{FRET}$. As shown in Supplementary Fig. 6b, at 0 h after transfection, the fluorescence signal of Rhodamine was very strong while that of FITC was very weak, indicating that the energy of FITC was transferred to Rhodamine and CLAN42 was intact at this time. With the elongation of incubating time, the fluorescence signal of Rhodamine decreased while that of FITC increased, indicating that the distance between Rhodamine and FITC increased and CLAN42 was broken or degraded. From these results, we can conclude that CLAN42 was gradually broken or degraded within 24 h after internalization by macrophages.

In addition, we detected the cytotoxicity of our CLAN42 by MTT assay. As shown in Supplementary Fig. 7, the cell viability of BMDMs incubated with CLAN42 (96.1%) and CLAN$_{mCas9/gRNA}$ (97.3%) was similar to that of PBS control, indicating that our CLAN42 was not cytotoxic.

**CLAN$_{mCas9/gNLRP3}$ disrupts NLRP3 of macrophages in vitro.** Before detecting the knockout efficiency of CLAN$_{mCas9/gNLRP3}$, we first analyzed Cas9 expression after CLAN$_{mCas9/gNLRP3}$ transfection. As demonstrated in Fig. 2a, Cas9 was efficiently expressed in BMDMs 12 and 24 h after CLAN$_{mCas9/gNLRP3}$ transfection, and the amount of protein expressed was correlated with the dose of Cas9 mRNA and time after transfection. Additionally, mCas9/gNLRP3 transfected by Lipofectamine MessengerMAX (Lipo$_{mCas9/gNLRP3}$) and pX330/gNLRP3 plasmid transfected with Lipofectamine 3000 (Lipo$_{pCas9/gNLRP3}$) also showed Cas9 expression, and Cas9 expression of Lipo$_{mCas9/gNLRP3}$ was higher than that in Lipo$_{pCas9/gNLRP3}$ within 24 h. By contrast, BMDMs transfected with phosphate-buffered saline (PBS) or free mCas9/gNLRP3 (Free$_{mCas9/gNLRP3}$) showed no Cas9 expression. Next, we detected the knockout efficiency by the T7 endonuclease I (T7E1) assay. The insertion/deletion (indel) frequency in the NLRP3 locus of BMDMs transfected with CLAN$_{mCas9/gNLRP3}$ at the dose of 0.85 nM was 16.3% at 12 h and increased to 35.6% at 24 h (Fig. 2b). The indel frequency of CLAN$_{mCas9/gNLRP3}$ was also dose-dependent and reached 70.2% at 24 h for the dose of 2.6 nM. The representative sequences of indels in the NLRP3 locus of BMDMs transfected with CLAN$_{mCas9/gNLRP3}$ are shown in Fig. 2c and Supplementary Fig. 8. Furthermore, the knockout efficiency was confirmed by detecting the protein expression of NLRP3. As shown in Fig. 2d, NLRP3 expression was reduced with CLAN$_{mCas9/gNLRP3}$ transfection, and the reduction was also dose dependent. Meanwhile, the expression of three other inflammasome proteins, NLRP1, NLRC4 and AIM2, were not affected by the knockout of NLRP3.

**CLAN$_{mCas9/gNLRP3}$ inhibits NLRP3 inflammasome activation.** We next tested whether CLAN$_{mCas9/gNLRP3}$-mediated NLRP3 knockout could inhibit the activation of the NLRP3 inflammasome. We transfected BMDMs with CLAN$_{mCas9/gNLRP3}$, and then stimulated BMDMs with the NLRP3 inflammasome stimulus nigericin after LPS priming[8]. As shown in Fig. 3a, compared with the Free$_{mCas9/gNLRP3}$ or CLAN$_{mCas9/gNC}$ group, CLAN$_{mCas9/gNLRP3}$ transfection dose-dependently inhibited the release of IL-1β into the supernatants (Sup.) of BMDMs. The amount of caspase-1 p10 (Casp-1 p10, an auto-cleaved fragment of caspase-1) was also dose-dependently reduced in the supernatants. The knockout efficiency of NLRP3 was confirmed by detecting NLRP3 protein expression in the cell lysates (Lys.) of BMDMs'. These results suggested that CLAN$_{mCas9/gNLRP3}$ inhibited NLRP3 inflammasome activation via disrupting NLRP3. Moreover, the inhibition efficiency of CLAN$_{mCas9/gNLRP3}$ at a dose of 2.6 nM was similar to that of glyburide (NLRP3 inflammasome inhibitor)[23]. Expression of IL-1β precursor (pro-IL-1β) and caspase-1 precursor (Casp-1 p45) was not affected by CLAN$_{mCas9/gNLRP3}$.

Because ASC oligomerization is a key event in NLRP3 inflammasome activation[3,4,18], we detected the formation of ASC oligomers in cross-linked cytosolic pellets of BMDMs. As displayed in Fig. 3b, ASC monomers (22 kDa) and higher order oligomers in cross-linked cytosolic pellets were attenuated by CLAN$_{mCas9/gNLRP3}$ transfection or glyburide treatment. The

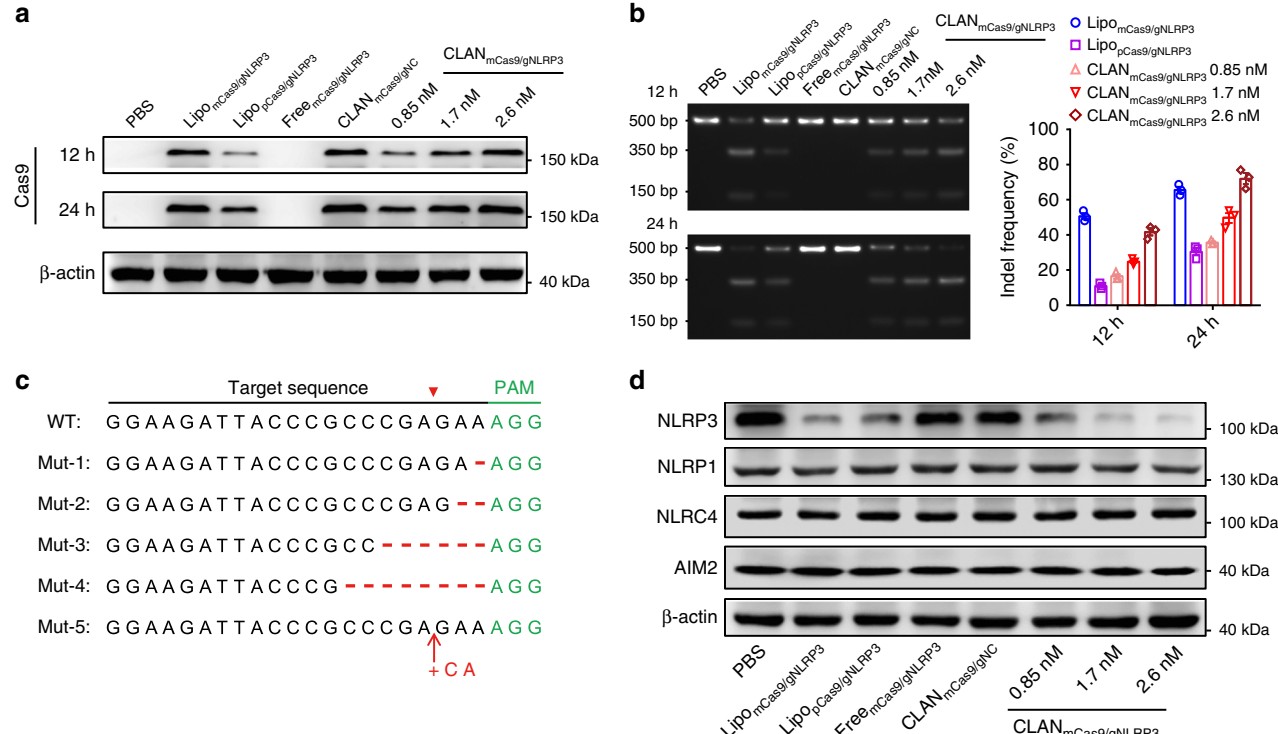

**Fig. 2** Detection of Cas9 expression and NLRP3 knockout efficiency in BMDMs transfected with CLAN$_{mCas9/gNLRP3}$ in vitro. **a**, **b** Immunoblot analysis of Cas9 protein expression (**a**) or T7E1 assay of indels introduced into the NLRP3 locus (**b**) in BMDMs transfected with different formulations at 12 or 24 h post-transfection. **c** Representative sequences of indels in the NLRP3 locus of BMDMs transfected with CLAN$_{mCas9/gNLRP3}$. See also Supplementary Fig. 8. WT, wild type, Mut, mutation. The PAM motif is in green, and the inserted bases are in red. **d** Immunoblot analysis of the NLRP3 knockout efficiency and other inflammasome protein expression in BMDMs transfected with different formulations at 72 h post-transfection. The data are representative of three independent experiments (**a**, **d**) or are shown as the means ± SEM of $n = 3$ (**b**)

difference was that CLAN$_{mCas9/gNLRP3}$ inhibited ASC oligomerization by reducing NLRP3 expression while glyburide suppressed NLRP3-ASC interaction[23]. In addition, the expression of ASC monomers in the cell lysates was correspondingly increased with the inhibition of ASC oligomerization after CLAN$_{mCas9/gNLRP3}$ transfection or glyburide treatment (Supplementary Fig. 9). Furthermore, we measured the release of IL-1β and IL-18 (another released cytokine of NLRP3 inflammasome activation) with ELISA. The release of IL-1β and IL-18 were dose-dependently reduced by CLAN$_{mCas9/gNLRP3}$ transfection (Fig. 3c–d), but TNF-α secretion, which was not dependent on the NLRP3 inflammasome[56], was not significantly affected by CLAN$_{mCas9/gNLRP3}$ and glyburide (Fig. 3e).

Subsequently, we tested whether NLRP3 inflammasome activation induced by other stimuli could be inhibited by CLAN$_{mCas9/gNLRP3}$-mediated NLRP3 knockout. The results showed that IL-1β secretion and caspase-1 cleavage induced by diverse stimuli (including MSU, nigericin, ATP and silica) were all inhibited by CLAN$_{mCas9/gNLRP3}$-mediated NLRP3 knockout (Fig. 3f, g)[8]. These results suggested that CLAN$_{mCas9/gNLRP3}$-mediated NLRP3 knockout is a potent strategy for the treatment of inflammatory diseases caused by diverse stimuli.

**CLAN$_{mCas9/gNLRP3}$ mediates NLRP3 knockout in vivo.** Before applying CLAN$_{mCas9/gNLRP3}$ to treat NLRP3-associated inflammatory diseases, we detected its NLRP3 knockout efficiency in vivo. C57BL/6 mice were injected with CLAN$_{mCas9/gNLRP3}$ at the dose of 0.5, 1, or 2 mg total RNA per kg body weight (0.5, 1, or 2 mg kg$^{-1}$). Mice injected with PBS, Free$_{mCas9/gNLRP3}$ (2 mg kg$^{-1}$) or CLAN$_{mCas9/gNC}$ (2 mg kg$^{-1}$) were used as control. Similar to the in vitro results, Cas9 protein was dose-dependently expressed

in peritoneal macrophages at 12 or 24 h after CLAN$_{mCas9/gNLRP3}$ injection (Fig. 4a). Next, we measured the NLRP3 knockout efficiency of CLAN$_{mCas9/gNLRP3}$ by detecting indels in the targeting locus of peritoneal macrophages using the T7E1 assay. The data showed that the indel frequency was dose-dependent and time-dependent, and reached 47.1% at 24 h after CLAN$_{mCas9/gNLRP3}$ (2 mg kg$^{-1}$) injection (Fig. 4b). The representative sequences of indels in the NLRP3 locus of peritoneal macrophages from mice injected with CLAN$_{mCas9/gNLRP3}$ (2 mg kg$^{-1}$) are displayed in Fig. 4c and Supplementary Fig. 10. Expression of the NLRP3 protein in peritoneal macrophages was also analyzed and showed that NLRP3 expression was dose-dependently inhibited by CLAN$_{mCas9/gNLRP3}$, while other inflammasome proteins, including NLRP1, NLRC4 and AIM2, were not affected (Fig. 4d). The expression of NLRP3 in CLAN$_{mCas9/gNLRP3}$ groups at the doses of 0.5, 1, and 2 mg kg$^{-1}$ was 81.5, 64.3, and 56.8% of that in PBS group, respectively (Fig. 4d). Considering that mice were only injected with CLAN$_{mCas9/gNLRP3}$ for a single injection, the knockout efficiency was acceptable. Because off-target effects are a large concern for the clinical application of CRISPR/Cas9[57], we isolated the genome of peritoneal macrophages at 72 h after CLAN$_{mCas9/gNLRP3}$ (2 mg kg$^{-1}$) injection and detected the indel frequency in potential off-target sites of NLRP3 gRNA. As shown in Supplementary Fig. 11, the indel frequency in six potential off-target sites was lower than 0.5%, while the indel frequency in the NLRP3 locus was 58.6%. These results indicated that CLAN$_{mCas9/gNLRP3}$ could disrupt the NLRP3 gene of macrophages in vivo with a low off-target effect.

**CLAN$_{mCas9/gNLRP3}$ mitigates LPS-induced septic shock.** We have demonstrated that CLAN$_{mCas9/gNLRP3}$ can inhibit NLRP3

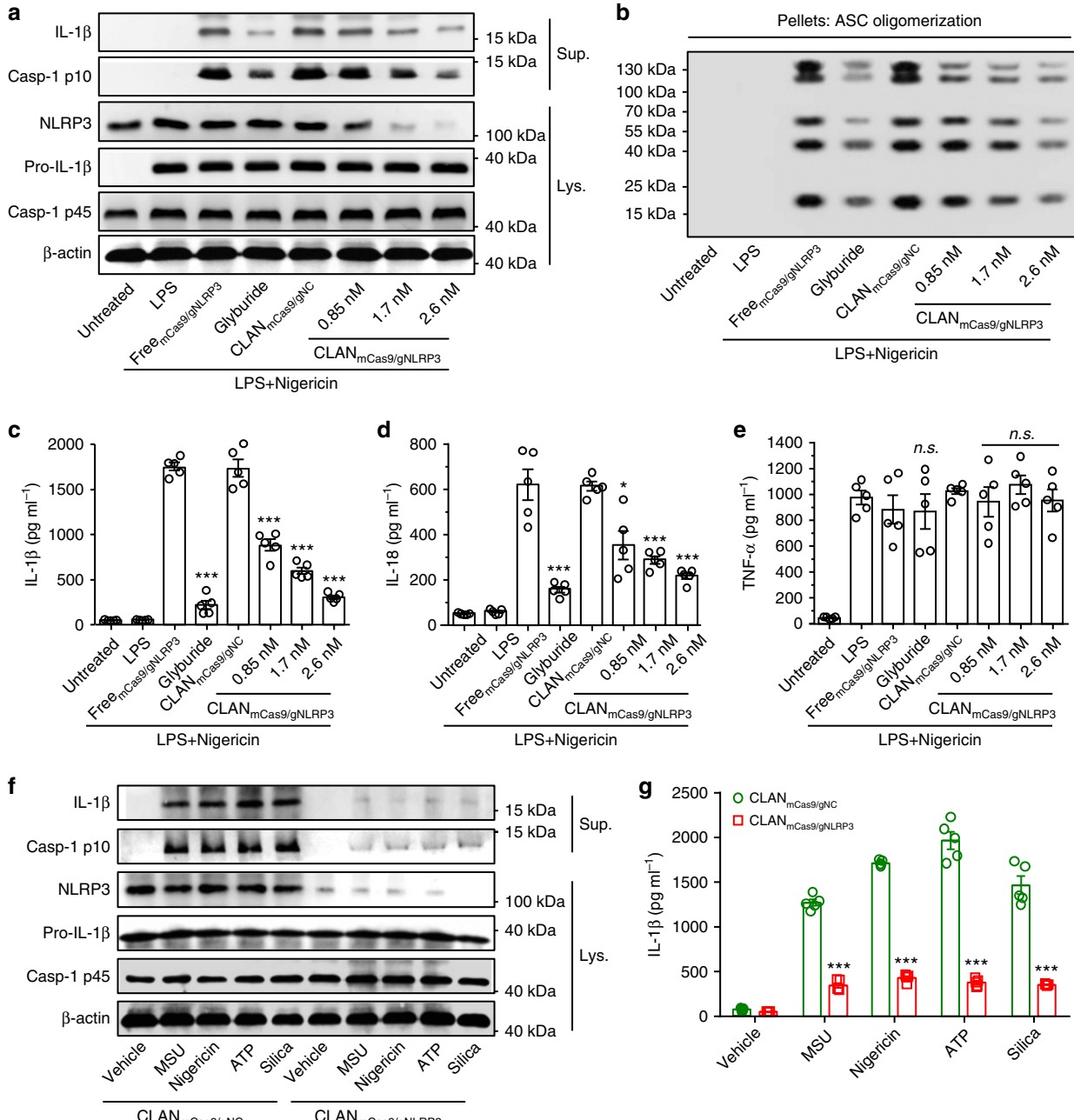

**Fig. 3** CLAN$_{mCas9/gNLRP3}$-mediated NLRP3 knockout and inflammasome inactivation in vitro. **a, b** Immunoblot analysis of cell lysates (Lys.) and supernatants (Sup.) (**a**) or cross-linked cytosolic pellets (**b**) from BMDMs transfected with CLAN$_{mCas9/gNLRP3}$ or other formulations and then primed with LPS and stimulated with nigericin. **c–e** ELISA of IL-1β (**c**), IL-18 (**d**), TNF-α (**e**) secretion from BMDMs transfected with CLAN$_{mCas9/gNLRP3}$ or other formulations and then primed with LPS and stimulated with nigericin. **f, g** Immunoblot analysis of cell lysates and supernatants (**f**) or ELISA of IL-1β secretion (**g**) from BMDMs transfected with CLAN$_{mCas9/gNLRP3}$ or CLAN$_{mCas9/gNC}$ at the dose of 2.6 nM mCas9 and then primed with LPS and stimulated with MSU, nigericin, ATP or silica. The data are representative of three independent experiments (**a, b, f**) or are shown as the means ± SEM of $n = 5$ (**c–e, g**). Two-way ANOVA (**c–e**) Student's $t$-test (**g**) *$P < 0.05$, ***$P < 0.001$, *n.s.* not significant

inflammasome activation via disrupting NLRP3 of macrophages. We next investigated whether CLAN$_{mCas9/gNLRP3}$ could be used to treat inflammatory diseases. It was reported that LPS-induced septic shock is NLRP3 inflammasome dependent, and the inhibition of its activation could mitigate septic shock[8,9]. Thus, we pre-treated mice with CLAN$_{mCas9/gNLRP3}$ and then challenged the mice with LPS to induce septic shock (Fig. 5a). As shown in Fig. 5a, although nearly 90% of mice treated with PBS, Free$_{mCas9/gNLRP3}$ or CLAN$_{mCas9/gNC}$ died within 30 h after LPS injection,

and all mice in the CLAN$_{mCas9/gNLRP3}$-treated groups (1, 2, and 4 mg kg$^{-1}$) were alive at this time point (log-rank test: $P < 0.001$). The median survival durations for the CLAN$_{mCas9/gNLRP3}$-treated groups at the doses of 1, 2, and 4 mg kg$^{-1}$ were 35 h, 37.5 h, and 42.5 h, respectively, which were longer than those of the PBS, Free$_{mCas9/gNLRP3}$ and CLAN$_{mCas9/gNC}$ groups (25 h, 26.5 h, and 24.5 h, respectively).

To confirm CLAN$_{mCas9/gNLRP3}$ mitigated LPS-induced septic shock via inhibiting NLRP3 inflammasome activation, we isolated

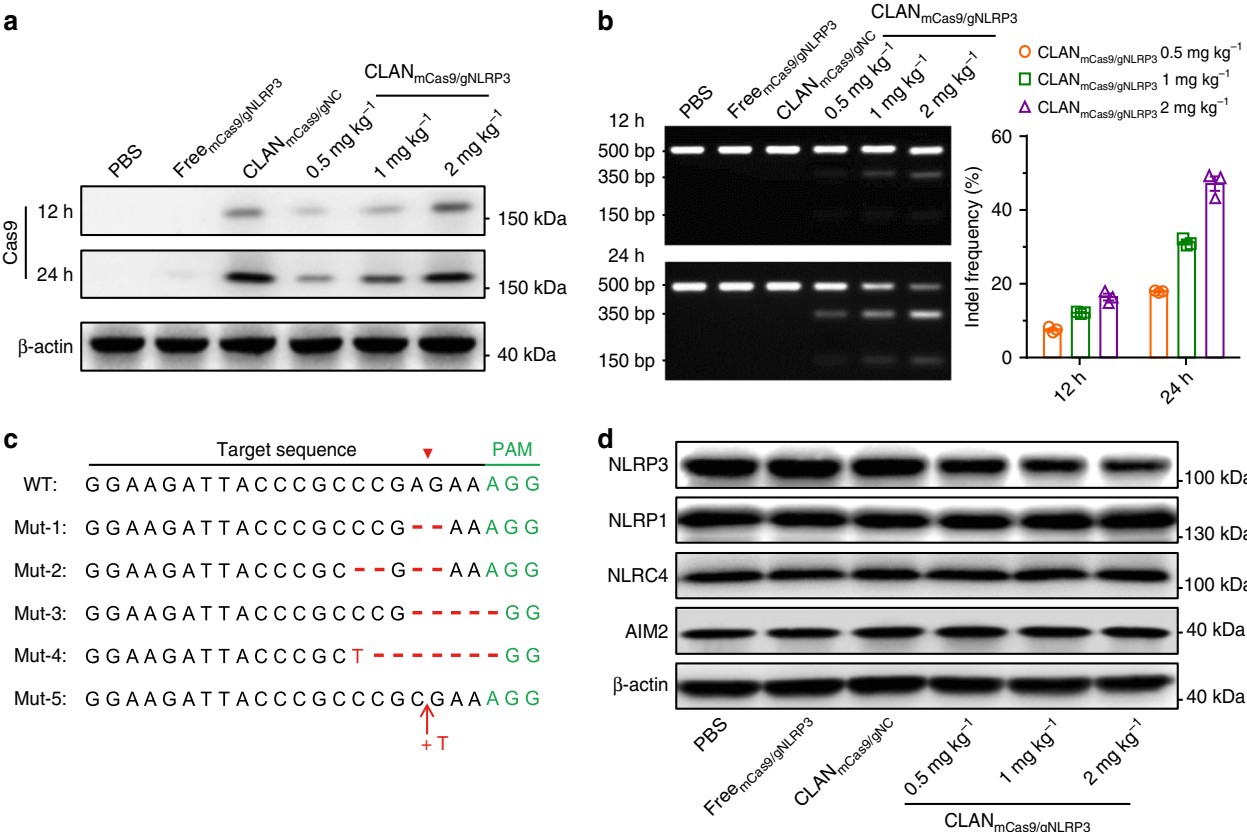

**Fig. 4** Detection of Cas9 expression and NLRP3 knockout efficiency in macrophages of mice injected with CLAN$_{mCas9/gNLRP3}$. **a**, **b** Immunoblot analysis of Cas9 protein expression (**a**) or the T7E1 assay of indels introduced into the NLRP3 locus (**b**) in peritoneal macrophages of mice injected with CLAN$_{mCas9/gNLRP3}$ at 12 or 24 h post-injection. $n = 5$ per group. **c** Representative sequences of indels in the NLRP3 locus, see also Supplementary Fig. 10. **d** Immunoblot analysis of NLRP3 and other inflammasome protein expression in peritoneal macrophages of mice injected with CLAN$_{mCas9/gNLRP3}$ at 72 h post-injection. $n = 5$ per group. The data are representative of two independent experiments (**a**, **d**) or are shown as the means ± SEM of $n = 3$ (**b**)

serum and peritoneal macrophages of mice at 4 h post LPS injection. The results showed that CLAN$_{mCas9/gNLRP3}$ dose-dependently inhibited the production of serum IL-1β and caspase-1 p10, which was in accordance with the reduction of NLRP3 expression in peritoneal macrophages (Fig. 5b). ELISA revealed that the production of IL-1β and IL-18 in the serum were inhibited by CLAN$_{mCas9/gNLRP3}$ (Fig. 5c, d). In addition, CLAN$_{mCas9/gNLRP3}$ had only a mild effect on the production of serum TNF-α (Fig. 5e). These results suggested that CLAN$_{mCas9/gNLRP3}$ could inhibit NLRP3 inflammasome activation in vivo via NLRP3 knockout and could mitigate LPS-induced septic shock.

In addition, we compared the efficacy of our CLAN$_{mCas9/gNLRP3}$ with the specific inhibitor of NLRP3 inflammasome (MCC950)[18]. As shown in Supplementary Fig. 12a, compared with PBS group, both CLAN$_{mCas9/gNLRP3}$ and MCC950 treatments extended the survival of LPS-challenged mice. The survival of mice treated with CLAN$_{mCas9/gNLRP3}$ was longer than that of mice treated with MCC950 (log-rank test: $P = 0.028$). The median survival duration for the CLAN$_{mCas9/gNLRP3}$-treated group was 44 h while that of the MCC950-treated group was 38 h. The secretion of IL-1β was inhibited by the treatments of CLAN$_{mCas9/gNLRP3}$ and MCC950, and CLAN$_{mCas9/gNLRP3}$ was more effective than MCC950 (Supplementary Fig. 12b). The reason was that the half-life of MCC950 is very short while CLAN$_{mCas9/gNLRP3}$-mediated NLRP3 knockout can last for a longer period of time as long as the edited macrophages are still alive[18].

**CLAN$_{mCas9/gNLRP3}$ mitigates MSU-induced peritonitis**. Because we have demonstrated that CLAN$_{mCas9/gNLRP3}$ was capable of mitigating LPS-induced systemic inflammation, we further studied the protective effect of CLAN$_{mCas9/gNLRP3}$ on periphery inflammation. Similarly, we pre-treated mice with CLAN$_{mCas9/gNLRP3}$ and then challenged the mice with MSU to induce peritonitis (Fig. 6a), which is also NLRP3 inflammasome dependent[8,10]. Peritoneal lavage fluid of the mice was collected 6 h after MSU injection. As shown in Fig. 6a, intraperitoneal administration of MSU increased the percentage of neutrophils (CD11b$^+$Ly6G$^+$) in the peritoneal cavity (from 4.48% to more than 80%), and CLAN$_{mCas9/gNLRP3}$ treatment could dose-dependently reduce the recruitment of neutrophils. Correspondingly, the total number of neutrophils accumulated in the peritoneal cavity was also reduced by CLAN$_{mCas9/gNLRP3}$ (Fig. 6b). Moreover, we detected the production of IL-1β in the peritoneal lavage fluid and NLRP3 expression in peritoneal macrophages. The results showed that CLAN$_{mCas9/gNLRP3}$ treatment inhibited MSU-induced IL-1β production in a dose-dependent manner (Fig. 6c) in accordance with the knockout efficiency of NLRP3 (Fig. 6d). From these results, we can conclude that CLAN$_{mCas9/gNLRP3}$-mediated NLRP3 knockout in vivo can be used to attenuate systemic and peritoneal inflammation by inhibiting NLRP3 inflammasome activation.

**Amelioration of HFD-induced T2D with CLAN$_{mCas9/gNLRP3}$**. The T2D incidence has greatly increased in recent years and has

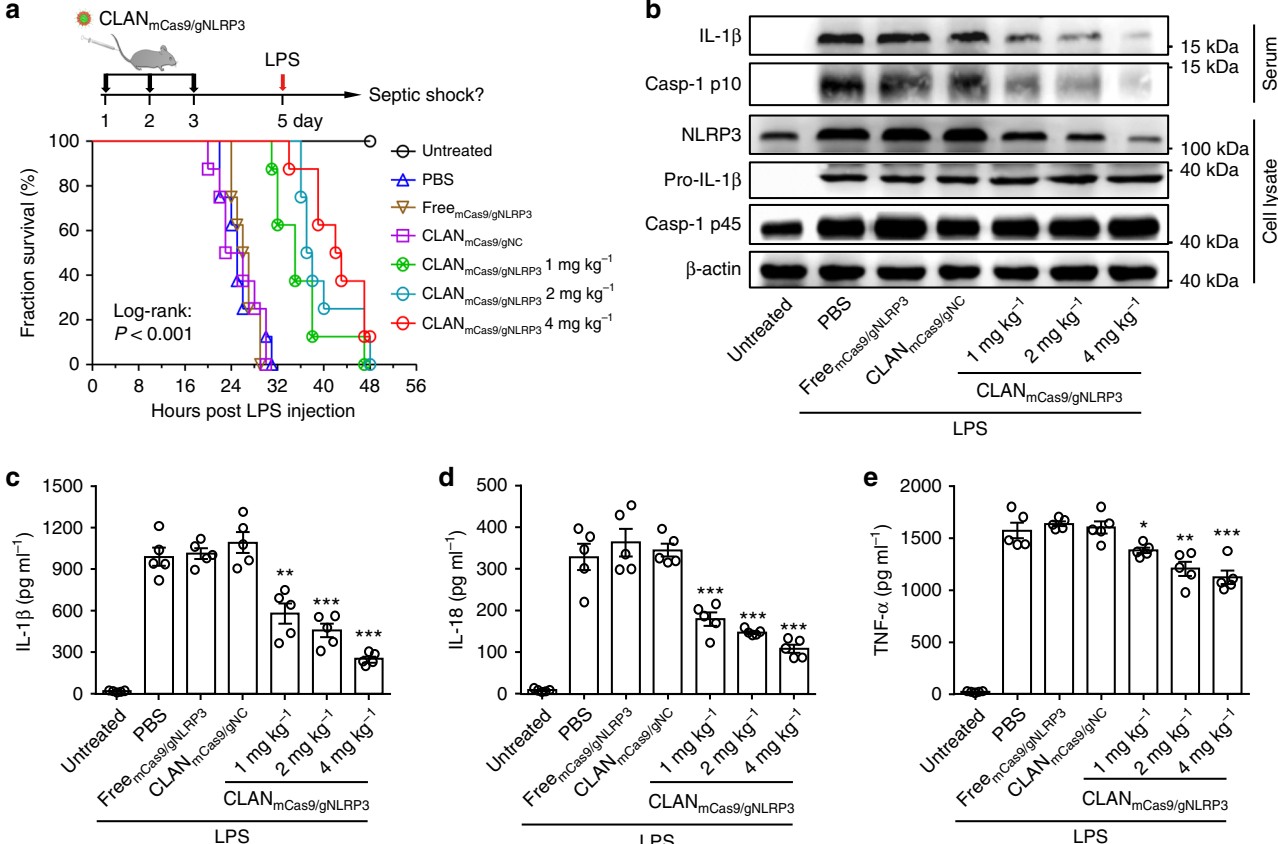

**Fig. 5** Mitigation of LPS-induced septic shock via CLAN$_{mCas9/gNLRP3}$-mediated NLRP3 knockout. **a** Therapy scheme and mouse survival curve of LPS-induced septic shock. $n = 8$ per group. **b–e** Immunoblot analysis of serum and peritoneal macrophage lysates (**b**) or ELISA of the IL-1β (**c**), IL-18 (**d**), TNF-α (**e**) concentration in serum from mice pre-treated with CLAN$_{mCas9/gNLRP3}$ or other formulations and then challenged with LPS. The data are representative of two independent experiments (**b**) or are shown as the means ± SEM of $n = 5$ (**c–e**). Log-rank test (**a**) Two-way ANOVA (**c–e**) *$P < 0.05$, **$P < 0.01$, ***$P < 0.001$

become a large threat to health worldwide. Previous studies have shown that inflammation is a key contributor to insulin resistance and T2D, and the NLRP3 inflammasome plays a critical role in the development of insulin resistance in T2D[14–16]. We thus investigated whether CLAN$_{mCas9/gNLRP3}$ can treat T2D by intravenously injecting HFD-induced T2D mice with CLAN$_{mCas9/gNLRP3}$ (Fig. 7a). As shown in Fig. 7b, the plasma glucose in the fasted states was dose-dependently reduced by CLAN$_{mCas9/gNLRP3}$ treatment, indicating that CLAN$_{mCas9/gNLRP3}$ ameliorated HFD-induced insulin resistance. To confirm the treatment effects of CLAN$_{mCas9/gNLRP3}$, we performed the glucose tolerance test (GTT) and insulin tolerance test (ITT) in HFD-induced T2D mice with different treatments. As shown in Fig. 7c, the glucose tolerance of T2D mice treated with CLAN$_{mCas9/gNLRP3}$ was superior to that of T2D mice treated with PBS, Free$_{mCas9/gNLRP3}$ or CLAN$_{mCas9/gNC}$. CLAN$_{mCas9/gNLRP3}$ (2 mg kg$^{-1}$) treatment showed similar improvement with the glyburide (a commercialized NLRP3 inflammasome inhibitor for T2D therapy) group[23]. Additionally, ITT also showed consistent results such that insulin sensitivity was enhanced after CLAN$_{mCas9/gNLRP3}$ treatment (Fig. 7d). The reduction in plasma glucose after insulin administration in the CLAN$_{mCas9/gNLRP3}$ group was similar to that in the glyburide group (Fig. 7d). The improvement of CLAN$_{mCas9/gNLRP3}$ treatment on glucose tolerance and insulin sensitivity also displayed a dose-dependent effect (Supplementary Fig. 13).

To further confirm that CLAN$_{mCas9/gNLRP3}$ improved the symptoms of HFD-induced T2D through NLRP3 knockout-mediated inhibition of the NLRP3 inflammasome, we detected

the expression of IL-1β, cleaved caspase-1 p10 and NLRP3 in white adipose tissue (WAT) of mice with different treatments. Compared with T2D mice treated with PBS, Free$_{mCas9/gNLRP3}$ or CLAN$_{mCas9/gNC}$, IL-1β secretion and caspase-1 cleavage in WAT was dose-dependently reduced by CLAM$_{mCas9/gNLRP3}$ treatment (Fig. 7e). In addition, the inhibition of NLRP3 inflammasome activation was consistent with the NLRP3 knockout efficiency of CLAN$_{mCas9/gNLRP3}$ treatments (Fig. 7e). The secretion of IL-1β and IL-18 in WAT as measured by ELISA also indicated dose-dependent inhibition of the NLRP3 inflammasome with CLAN$_{mCas9/gNLRP3}$ treatments (Fig. 7f, g). As shown in Fig. 7e–g, the secretion of IL-1β and IL-18 was also inhibited in glyburide-treated T2D mice, indicating that the activation of NLRP3 inflammasome was inhibited by glyburide in vivo, although glyburide didn't directly inhibit NLRP3 protein[23]. As to the improvement of T2D symptoms achieved by glyburide, it was not only due to its inhibition on NLRP3 inflammasome, but also may because of the effects of cyclohexylurea group on insulin secretion[23]. Furthermore, the secretion of two other proinflammatory cytokines, TNF-α and MCP-1, was dose-dependently inhibited by CLAN$_{mCas9/gNLRP3}$ treatments, suggesting that CLAN$_{mCas9/gNLRP3}$-mediated NLRP3 knockout could improve the overall inflammatory environment in WAT (Fig. 7h, i). In summary, CLAN$_{mCas9/gNLRP3}$-mediated NLRP3 knockout can improve the glucose tolerance and insulin sensitivity of HFD-induced T2D and ameliorate the inflammation of adipose tissue through inhibition of NLRP3 inflammasome activation.

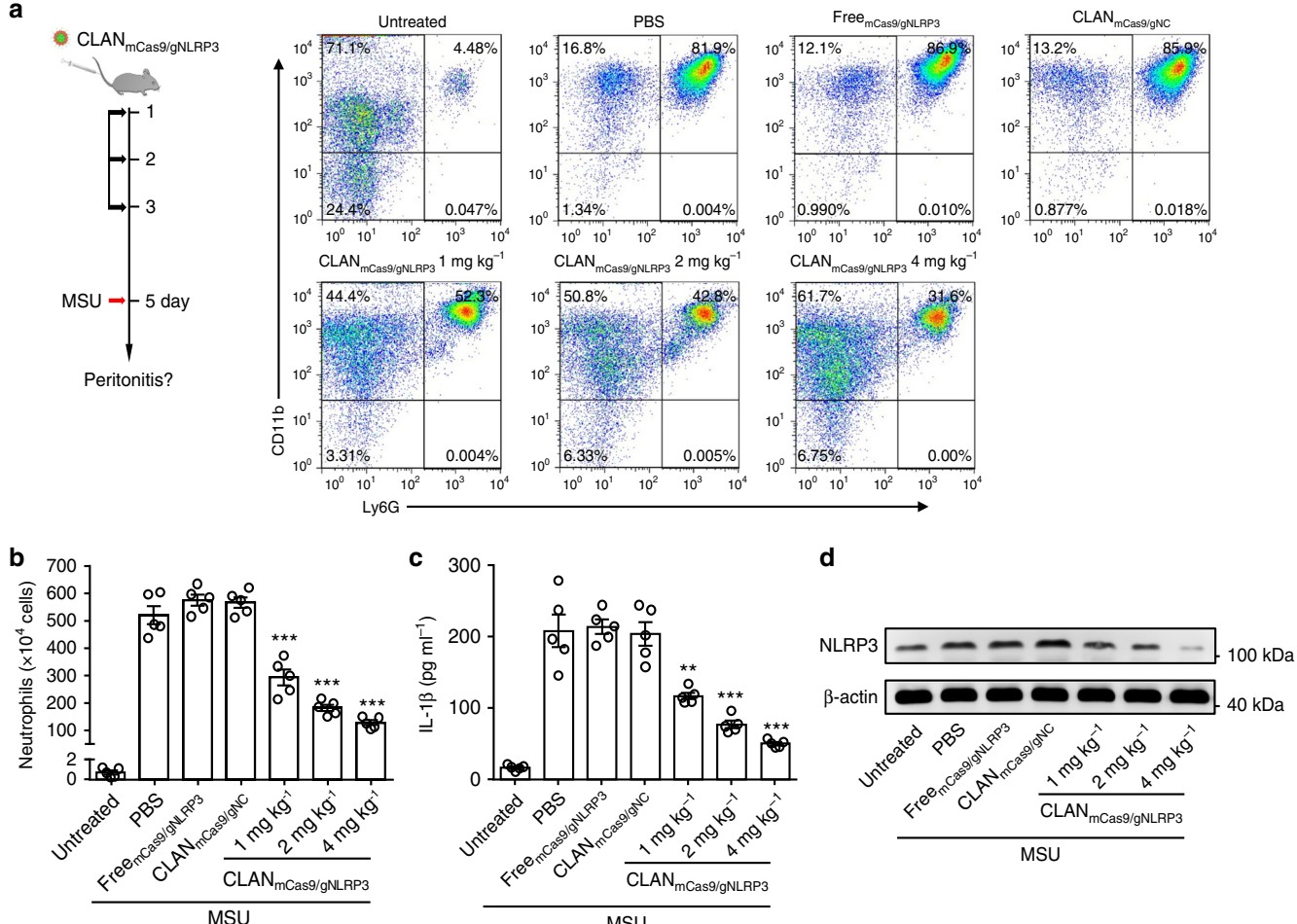

**Fig. 6** Mitigation of MSU-induced peritonitis via CLAN$_{mCas9/gNLRP3}$-mediated NLRP3 knockout. **a** Therapy scheme and FACS analysis of neutrophil recruitment in the peritoneal cavity of MSU-induced peritonitis. $n = 5$ per group. **b**–**d** Total number of neutrophils infiltrated in the peritoneal cavity (**b**), ELISA of IL-1β production in peritoneal lavage fluid (**c**), or immunoblot analysis of NLRP3 knockout efficiency in peritoneal macrophages (**d**) of mice pre-treated with CLAN$_{mCas9/gNLRP3}$ or other formulations and then challenged with MSU. The data are representative of two independent experiments (**a**, **d**) or are shown as the means ± SEM of $n = 5$ (**b**, **c**). Two-way ANOVA (**b**, **c**) **$P < 0.01$, ***$P < 0.001$

## Discussion

In this study, we selected the NLRP3 inflammasome as the therapeutic target, which is a major mediator of acute/chronic inflammation and has been proven to be an effective target for the treatment of multiple inflammatory diseases[8–19]. Nevertheless, current therapeutics targeting NLRP3 inflammasome, including inhibitors of upstream signals of the NLRP3 inflammasome or biologics targeting IL-1β or its receptors, either indirectly inhibit NLRP3 or only partially inhibit the signal of the NLRP3 inflammasome, limiting their therapeutic efficiency[20,24–26]. The CRISPR/Cas9 genome editing tool, which is of high efficiency, high specificity and simplicity[29], provides a good strategy for directly targeting NLRP3 and completely ablating NLRP3-dependent inflammation. The only barrier for applying CRISPR/Cas9 to the treatment of inflammatory diseases is how to efficiently deliver Cas9 and gRNA into immune cells, especially macrophages[58]. Several non-viral delivery systems have been used to deliver the Cas9/gRNA ribonucleoprotein complex, mRNA/gRNA or plasmids into hepatocytes, tumor cells or some tissue cells in vivo; no delivery system targeting macrophages have been reported[59,60].

Our study has screened the polymer nanoparticle CLAN with optimized properties for delivering Cas9 mRNA and gRNA into macrophages for gene editing. The results showed that, within a certain range, CLAN with a higher surface charge and lower PEG density could be internalized by macrophages more efficiently[54,55]. Increasing the surface charge was more effective than reducing the PEG density. Because mRNA and gNLRP3 can be rapidly degraded after gene editing, genomic ablation of NLRP3 would not cause integration mutations of the genome. We have demonstrated that screened CLAN could encapsulate mCas9/gNLRP3 and protect mCas9/gNLRP3 from RNase of the physiological environment. We next demonstrated that CLAN$_{mCas9/gNLRP3}$ transfection could express Cas9 protein within several hours and disrupt the NLRP3 gene in BMDMs, inhibiting the activation of the NLRP3 inflammasome in response to diverse stimuli. The secretion of both proinflammatory cytokines IL-1β and IL-18 was reduced by CLAN$_{mCas9/gNLRP3}$. We noted that CLAN$_{mCas9/gNLRP3}$ did not completely inactivate the NLRP3 inflammasome; thus, it is possible that the transfection efficiency was not 100%. Future work will aim to enhance other properties of CLAN to increase its transfection efficiency.

Furthermore, we have demonstrated that CLAN$_{mCas9/gNLRP3}$ was capable of dose-dependently disrupting NLRP3 of macrophages in vivo. Because off-target effects are a huge concern for the clinical application of CRISPR/Cas9[57], we detected the indel frequency in the predicted off-target sites after the intravenous injection of CLAN$_{mCas9/gNLRP3}$. Our results showed that only a

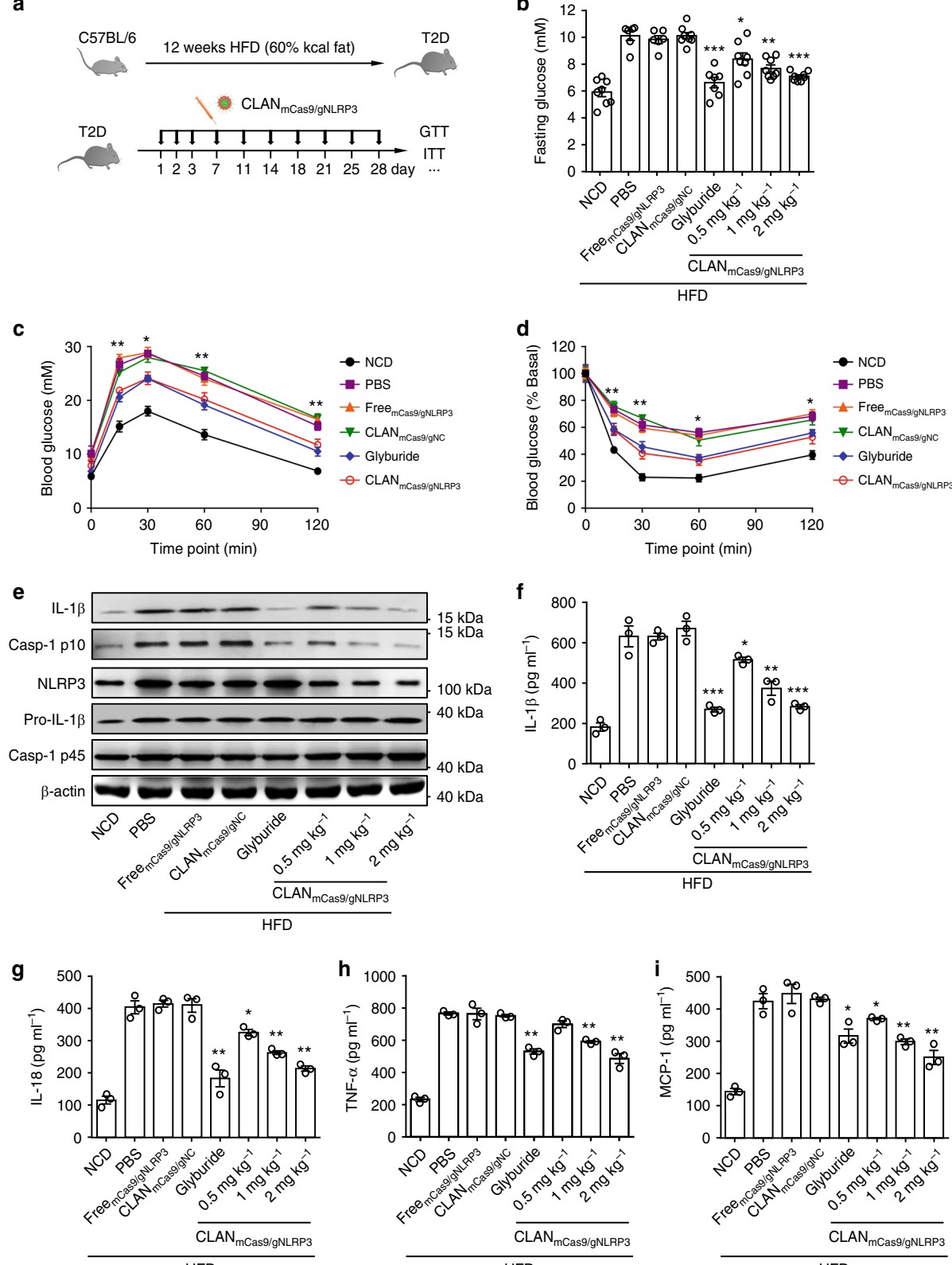

**Fig. 7** Treatment of HFD-induced T2D via CLAN$_{mCas9/gNLRP3}$-mediated NLRP3 knockout. **a** Scheme of treatment. Mice fed on a normal chow diet (NCD) were used as a healthy control. **b–d** Fasting glucose (**b**), GTT (**c**), or ITT (**d**) of T2D mice treated with CLAN$_{mCas9/gNLRP3}$ or other formulations, see also Supplementary Fig. 13, $n = 6$–8 per group. **e** Immunoblot analysis of WAT lysates from T2D mice treated with CLAN$_{mCas9/gNLRP3}$ or other formulations. WAT was collected after GTT and ITT measurement. **f–i** WAT of T2D mice treated with CLAN$_{mCas9/gNLRP3}$ or other formulations were cultured for 24 h, and supernatants were analyzed by ELISA to detect IL-1β (**f**), IL-18 (**g**), TNF-α (**h**), and MCP-1 (**i**) release from WAT. The data are shown as the means ± SEM of $n = 6$–8 (**b–d**) or $n = 3$ (**f–i**) or are representative of three independent experiments (**e**). Two-way ANOVA (**b–d**), **f–i** *$P < 0.05$, **$P < 0.01$, ***$P < 0.001$

low off-target effect was detected in macrophages. In addition, our CLAN was internalized by different immunocytes, including macrophages, neutrophils, T and B cells, but only macrophages and neutrophils, which have strong phagocytic activity, can effectively internalize it (Supplementary Fig. 14). Because NLRP3 only functions in macrophages, disruption of NLRP3 gene in other cells won't cause foreseeable side effects. And the life-span of macrophages ranges from 6 to 16 days, the disrupted NLRP3 of macrophages will recover in a short time and won't cause permanent NLRP3 inhibition. Thus, $CLAN_{mCas9/gNLRP3}$ has great potential for the clinical treatment of NLRP3-dependent inflammatory diseases. We have applied $CLAN_{mCas9/gNLRP3}$ for the prevention of LPS-induced septic shock and MSU-induced peritonitis, and it mitigated the acute inflammatory responses. $CLAN_{mCas9/gNLRP3}$ was unable to entirely prevent septic shock and peritonitis likely because of the activation of other inflammasomes, such as NLRP1, NLRC4, and AIM2[3,5,7], and the inactivation of the NLRP3 inflammasome was insufficient to inhibit acute inflammation. In the treatment of HFD-induced T2D, $CLAN_{mCas9/gNLRP3}$ reduced chronic inflammation in adipose tissue and improved insulin sensitivity via the knockout of NLRP3. Additionally, the dosing frequency of $CLAN_{mCas9/gNLRP3}$ was lower than that of commercialized glyburide, which is injected daily. Thus, $CLAN_{mCas9/gNLRP3}$ was an effective therapeutic for T2D, and its therapeutic effect was likely to last longer due to the genomic ablation of NLRP3.

In conclusion, we have developed a strategy to deliver Cas9 mRNA and gRNA into macrophages for gene editing. Delivery of mCas9/gNLRP3 with CLAN could mitigate acute and chronic inflammatory diseases. Our results have proven $CLAN_{mCas9/gNLRP3}$ to be a promising therapy for NLRP3-dependent inflammatory diseases. Our study also provided an example for treating immune-related diseases by nanoparticles-mediated gene editing of immune cells.

## Methods

**Materials and reagents**. N,N-Bis(2-hydroxyethyl)-N-methyl-N-(2-cholesteryoxycarbonyl-aminoethyl) ammonium bromide (BHEM-Chol), $PEG_{5K}$-b-$PLGA_{11K}$ (molar ratio: LA/GA = 75/25) and $PLGA_{11K}$ (molar ratio: LA/GA = 75/25) polymers were synthesized, purified and characterized as previously reported[61,62]. Cy5-siRNA containing a scrambled sequence (antisense strand, 5′-ACGUGACACGU UCGGAGAAdTdT-3′) was provided by Suzhou Ribo Life Science Co. Ltd. (Kunshan, China). mCas9 or mCas9-EGFP gRNAs were prepared by in vitro transcription (IVT) according to the manufacturer's instructions. mCas9 or mCas9-EGFP was prepared using the HiScribe™ T7 ARCA mRNA Kit (with tailing) (New England Biolabs, MA, USA) and PCR amplification products of pX330 or pX458 (addgene) as a template. gRNAs (gNLRP3, gGFP and gNC) were designed using the CRISPR.mit.edu platform, cloned into pX330 or pX458 (oligos were listed in Supplementary Table 3)[63] and then prepared using the HiScribe™ T7 Quick High Yield RNA Synthesis Kit (New England Biolabs, MA, USA). Primers for IVT of mRNA and gRNAs are listed in Supplementary Table 4. Anti-CD11b (PerCP/Cy5.5), anti-F4/80 (APC), and anti-Ly6G (PE) were purchased from Biolegend (CA, USA). Lipofectamine™ 3000, and Lipofectamine™ MessengerMAX™ reagent were purchased from Invitrogen (CA, USA). Ultrapure LPS and glyburide were obtained from Invivogen (CA, USA). MSU, nigericin, ATP, and silica were purchased from Sigma (MO, USA).

**Preparation of CLAN library and $CLAN_{mCas9/gRNA}$**. $CLAN_{Cy5-siRNA}$ were prepared by the double emulsion method according to our previous reports[61,62]. In brief, Cy5-siRNA (200 μg in 25 μl RNase free water) was emulsified in 500 μl of chloroform containing different amounts of BHEM-Chol, $PEG_{5K}$-b-$PLGA_{11K}$ and $PLGA_{11K}$ for 1 min at 80 W over an ice bath using Vibra-Cell VCX130 from Sonics & Materials, Inc. (Newtown, USA). Subsequently, 5 ml of RNase-free water was added, and the mixture was further emulsified before removing chloroform by a rotary evaporator. To prepare a CLAN library of different nano-properties, the weight of BHEM-Chol was adjusted to 1, 1.5, 2, and 3 mg, while the mass fraction of $PEG_{5K}$-b-$PLGA_{11K}$ in the polymer was adjusted to 100, 91.3, 86.4, and 81.4% (Supplementary Fig. 1). The accurate weight of $PEG_{5K}$-b-$PLGA_{11K}$ in 500 μl of chloroform was 25, 21.9, 20.3, and 18.8 mg, while the weight of $PLGA_{11K}$ was simultaneously changed to 0, 2.1, 3.2, and 4.3 mg (Supplementary Fig. 1). Similar to $CLAN_{Cy5-siRNA}$, $CLAN_{mCas9/gRNA}$ was also prepared by the double emulsion method by dissolving 125 μg of mCas9 and 75 μg of gRNA in 25 μl of RNase-free water, in which the molar ratio of mCas9/gRNA was 1/25.

**Characterization of $CLAN_{Cy5-siRNA}$ and $CLAN_{mCas9/gRNA}$**. The diameter and zeta potential of $CLAN_{Cy5-siRNA}$ and $CLAN_{mCas9/gRNA}$ were characterized using the Malvern Zetasizer Nano ZS90 system (Worcestershire, UK) as described previously[61]. The mass fraction of $PEG_{5K}$-b-$PLGA_{11K}$ incorporated in CLAN was detected by ${}^1H$ NMR spectroscopy. $CLAN_{Cy5-siRNA}$ and $CLAN_{mCas9/gRNA}$ were destructed by dimethylsulfoxide, and then Cy5-siRNA and mCas9/gRNA were displaced by 10 mg ml⁻¹ heparin sodium at 25 °C for 1 h, followed by extraction to measure the encapsulation efficiency. The encapsulation efficiency of Cy5-siRNA in $CLAN_{Cy5-siRNA}$ was measured by HPLC according to our previous report[61]. The encapsulation efficiency of mCas9 and gRNA in $CLAN_{mCas9/gRNA}$ was measured using the Quant-iT™ RiboGreen® RNA Reagent and Kit (Thermo Fisher Scientific, MA, USA) according to the manufacturer's protocol. The morphology of $CLAN_{mCas9/gRNA}$ was tested by JEOL-2010 microscopy (Tokyo, Japan) at an accelerating voltage of 200 kV. The colloidal stability of $CLAN_{mCas9/gRNA}$ was examined in PBS with 10% FBS at 37 °C, and the diameter was characterized at various incubation times. The integrity of mCas9 and gRNA in $CLAN_{mCas9/gRNA}$ was confirmed by 2% RNA denaturing gel electrophoresis after extracting mCas9/gRNA from $CLAN_{mCas9/gRNA}$.

**Mice**. C57BL/6 mice were purchased from Beijing HFK Bioscience (Beijing, China) and were housed in the specific pathogen-free facility in University of Science and Technology of China. At the end of experiments, all mice were killed by $CO_2$ inhalation. All animal experiments were approved by the Ethics Committee of the University of Science and Technology of China.

**Cell preparation**. Primary BMDMs were derived from the tibia and femoral bone marrow cells as described previously and were cultured in DMEM supplemented with 10% FBS (ExCell Bio Inc., Shanghai, China)[8,16]. Raw264.7-GFP cells were established by transfecting Raw264.7 cells (ATCC) with the CRISPR/Cas9 plasmid and GFP knockin donor plasmid to stably express GFP with β-actin. Monoclonal Raw264.7-GFP cells were amplified and maintained in DMEM supplemented with 10% FBS. Raw264.7 and Raw264.7-GFP cells were routinely tested for mycoplasma contamination.

**FACS analysis of CLAN with higher macrophage uptake**. $CLAN_{Cy5-siRNA}$ of different nano-properties was intravenously injected into C57BL/6 mice (6 weeks, n = 3 per group) at the Cy5-siRNA dose of 1 mg kg⁻¹ (i.e., 20 μg per mouse). Next, macrophages of the peritoneal cavity or immunocytes of the liver, peripheral blood, spleen, and adipose tissue were harvested 24 h after injection. Peritoneal macrophages were isolated by washing the peritoneal cavity with 10 ml of cold PBS. Liver and adipose tissue were digested with 2 mg ml⁻¹ collagenase I/II for 45 min at 37 °C, and then immunocytes were purified with 40% percoll (GE Healthcare, UK) according to the manufacturer's protocol. Spleen immunocytes were isolated by filtering the spleen through nylon-wool-glass beads to obtain a single-cell suspension. Subsequently, isolated macrophages or immunocytes were treated with red blood cell lysis buffer (Stemcell, Vancouver, Canada) and were stained with anti-CD11b and anti-F4/80 antibodies to label macrophages. The cellular uptake of $CLAN_{Cy5-siRNA}$ by macrophages was assessed by FACSVerse (BD, Bedford, USA). The percentage of Cy5-positive macrophages and the mean fluorescence intensity (MFI) of Cy5 in positive macrophages were analyzed with Flowjo 7.6.1. The relative uptake quantity of $CLAN_{Cy5-siRNA}$ was calculated by the following formula: Relative uptake quantity = (Percentage of Cy5 × MFI of Cy5)/100%.

**Selecting CLAN for mCas9/gRNA delivery**. To detect Cas9-EGFP mRNA expression or GFP knockout efficiency, BMDMs or Raw264.7-GFP cells were plated in 6-well plates (10⁶ cells per well) and were incubated overnight. Next, the medium was replaced, and cells were transfected with $CLAN_{mCas9-EGFP/gNC}$ at the final concentration of 2.6 nM mCas9-EGFP or $CLAN_{mCas9/gGFP}$ at the final concentration of 2.6 nM mCas9. Twenty-four hours after transfection, BMDMs were collected to detect EGFP expression by FACS. Transfected Raw264.7-GFP cells were cultured for four days before detecting the percentage of GFP-negative Raw264.7-GFP cells by FACS. To detect Cas9-EGFP mRNA expression in vivo, C57BL/6 mice (n = 3 per group) were intravenously injected with $CLAN_{mCas9-EGFP/gNC}$ at the total RNA dose of 2 mg kg⁻¹ (i.e., 1.25 mg of Cas9-EGFP mRNA per kg body weight). Twenty-four hours after injection, the mice were sacrificed, and peritoneal macrophages were collected to detect EGFP expression by FACS.

**Confocal microscope analysis**. To detect the intracellular localization of CLAN42, we firstly labeled mCas9 with Cy5 dye using Turbo Labeling™ Kit (Thermo Fisher Scientific, MA, USA). Then Cy5-mCas9 and gRNA were encapsulated into $CLAN_{Cy5-mCas9/gRNA}$. BMDMs were plated on glass coverslips overnight and incubated with $CLAN_{Cy5-mCas9/gRNA}$ for 6 h at the concentration of 2.6 nM Cy5-mCas9. At different time points (0, 6, 12, and 24 h) after transfection, BMDMs were collected, and stained with Alexa Fluor 488-labeled EEA1 antibody to mark endosome, Alexa Fluor 568-labeled Cas9 antibody to mark Cas9 protein and DAPI

to mark nucleus. The localization of CLAN42 and Cas9 protein was visualized by Ti-E A1 confocal microscope (Nikon, Tokyo, Japan).

To analyze the integrity of CLAN42, FITC and Rhodamine were selected as the donor and acceptor molecules of FRET couple, respectively. The emission spectrum of $CLAN_{FRET}$, $CLAN_{FITC}$, and $CLAN_{Rho}$ were detected with RF-6000 fluorescence spectrophotometer (Shimadzu, Tokyo, Japan) with the excitation light of 488 nm. Subsequently, we incubated BMDMs with $CLAN_{FRET}$ for 6 h at the concentration of 2.6 nM mCas9. At different time points (0, 6, 12, and 24 h) after transfection, BMDMs were collected, and stained with Alexa Fluor 647-labeled EEA1 antibody to mark endosome and DAPI to mark nucleus. The fluorescence of FITC and Rhodamine were observed with confocal microscope (Ti-E A1, Nikon, Tokyo, Japan) using the excitation light of 488 nm.

**BMDM transfection and stimulation.** BMDMs were plated in 6-well plates ($10^6$ cells per well) and were incubated overnight. Next, the medium was replaced, and cells were transfected with $CLAN_{mCas9/gNLRP3}$ or other formulations for 6 h. $CLAN_{mCas9/gNLRP3}$ was transfected at the dose of 0.85, 1.7, and 2.6 nM mCas9. $Lipo_{mCas9/gNLRP3}$ was transfected at the dose of 0.85 nM mCas9, while $Lipo_{pCas9/gNLRP3}$ was transfected at 0.23 nM of pX330/gNLRP3 plasmid—i.e., 1.25 µg ml$^{-1}$ for both mRNA and plasmid. $Free_{mCas9/gNLRP3}$ and $CLAN_{mCas9/gNC}$ were transfected at the dose of 2.6 nM mCas9 and was used as the negative control. To stimulate the activation of the NLRP3 inflammasome, 72 h after $CLAN_{mCas9/gNLRP3}$ transfection, cells were primed with 500 ng ml$^{-1}$ LPS for 3 h and then were stimulated with nigericin (10 µM) for 30 min or MSU (150 µg ml$^{-1}$) for 4 h or ATP (2.5 mM) for 30 min or silica (500 µg ml$^{-1}$) for 6 h. The supernatants and BMDMs were then collected for immunoblotting or ELISA or the ASC oligomerization assay. For glyburide, LPS-primed BMDMs were pretreated with glyburide (200 µM) for 30 min and then were stimulated with nigericin.

**Immunoblot analysis.** Cell lysates were prepared by lysing BMDMs and peritoneal macrophages in Laemmli sample buffer. The protein content of supernatants or serum was concentrated using StrataClean resin (Agilent, CA, USA). WAT proteins were extracted using the Total Protein Extraction Kit for Adipose Tissues (Invent, MN, USA). All protein samples were prepared according to the manufacturer's instructions, separated by 12% SDS-PAGE and transferred to PVDF membranes using a wet-transfer system. The membranes were blocked in TBST buffer containing 5% (wt/vol) BSA for 2 h at room temperature and then were incubated with primary antibody diluted in TBST buffer containing 1% (wt/vol) BSA overnight at 4 °C. The membranes were further incubated with HRP-conjugated secondary antibody diluted in TBST buffer containing 1% (wt/vol) BSA for 1 h at room temperature. Finally, the membranes were developed using SuperSignal Chemiluminescent Substrates (Thermo Fisher Scientific, MA USA). Uncropped scans of the most important blots were shown in Supplementary Fig. 15–18.

Primary antibodies: Anti-Cas9 (ab191468, 1:1000) and anti-NLRP1 (ab3683, 1:1000) antibodies were from Abcam (Shanghai, China). Anti-NLRP3 (AG-20B-0014, 1:1000) and anti-ASC (AL177, 1:1000) antibodies were from Adipogen (CA, USA). Anti-NLRC4 (06–1125, 1:1000) antibody was from EMD Millipore (MA, USA). Anti-AIM2 (sc-137967, 1:1000), anti-caspase-1 p10 (sc-514, 1:1000), and anti-β-actin antibodies were from Santa Cruz (CA, USA). Anti-IL1β (AF-401-NA, 1:1000) was from R&D (MN, USA). Secondary HRP-conjugated antibodies anti-mouse IgG, anti-rabbit IgG and anti-goat IgG (all 1:1000) were from EMD Millipore (MA, USA).

**ASC oligomerization assay.** BMDMs were washed in cold PBS, and 500 µl of cold buffer (20 mM HEPES-KOH, pH 7.5, 150 mM KCL, 1% NP-40, 0.1 mM PMSF, 1 mg ml$^{-1}$ leupeptin, 11.5 mg ml$^{-1}$ aprotinin, and 1 mM sodium orthovanadate) was added. Cells were dissolved on the shaker for 30 min at 4 °C and then were centrifuged at $330 \times g$ for 10 min at 4 °C. The pellets were washed twice in 1 ml of cold PBS and were resuspended in 500 µl of cold PBS. Next, 2 mM disuccinimidyl suberate (Sigma, MO, USA) was added to the resuspended pellets, which were incubated for 30 min with rotation at room temperature. Samples were then centrifuged at $330 \times g$ for 10 min at 4 °C. The supernatants were removed, and the cross-linked pellets were resuspended in 30 µl of Laemmli sample buffer. Samples were boiled for 10 min at 99 °C and were analyzed by immunoblotting.

**T7E1 assay and Sanger sequencing.** Genomic DNA of BMDMs or peritoneal macrophages was extracted using the AxyPrep multisource genomic DNA miniprep kit according to the manufacturer's instructions. The flanking region of the gNLRP3 targeting sequence and its potential off-targets was amplified by PCR (On-target and off-targets of gNLRP3, as well as primers for PCR amplification, are listed in Supplementary Table 5–7). Two hundred nanograms of purified PCR products were first diluted in 20 µl of NEBuffer 2 and was denatured at 95 °C for 5 min. The denatured PCR products were then reannealed at 95–85 °C (ramp rate: −2 °C s$^{-1}$) and at 85–25 °C (ramp rate: −0.1 °C s$^{-1}$). After cooling down to 4 °C, 1 µl of T7E1 (NEB, Beijing, China) was added and incubated at 37 °C for 15 min. Finally, 1.5 µl of 0.25 M EDTA was added to stop the reaction. The fragmented PCR products were analyzed with 2% agarose gel electrophoresis, and the percent of nuclease-specific cleavage products (fraction cleaved) was determined by Image

J. The gene knockout efficiency was calculated using the following formula: Indel frequency (%) $= 100\% \times (1 - \sqrt{1 - \text{fraction cleaved}})$. The PCR products of the genomic region-flanking target sites of gNLRP3 were cloned into the T-clone vector for Sanger sequencing by General Biosystems (Anhui, China).

**ELISA.** Supernatants from cells or WAT culture, serum, and peritoneal lavage fluid were analyzed using IL-1β, IL-18, TNF-α, and MCP-1 ELISA kits according to the manufacturer's instructions. Mouse IL-1β/IL-1F2 ELISA kit (DY401) was from R&D (MN, USA). ELISA detection kits for mouse IL-18 (ELM-IL18), TNF-α (ELM-TNFa), and MCP-1 (ELM-MCP1) were from RayBiotech (GA, USA).

**Mitigation of LPS-induced septic shock.** C57BL/6 mice (6–8 weeks, $n = 8$ per group) were first injected with $CLAN_{mCas9/gNLRP3}$ at the dose of 1, 2, or 4 mg kg$^{-1}$ total RNA for three injections. Mice injected with PBS and $Free_{mCas9/gNLRP3}$ (4 mg kg$^{-1}$), $CLAN_{mCas9/gNC}$ (4 mg kg$^{-1}$) were used as negative controls. Mice intraperitoneally injected with MCC950 (50 mg kg$^{-1}$) were used as positive controls. Two days after the last $CLAN_{mCas9/gNLRP3}$ injection, all mice were intraperitoneally injected with LPS (20 mg kg$^{-1}$) to induce septic shock. Next, mice were further observed to construct a survival curve or were sacrificed at 4 h after LPS injection to collect serum and peritoneal macrophages. Serum was subjected to immunoblotting to detect IL-1β or caspase p10 or ELISA to measure the IL-1β, IL-18 and TNF-α concentrations. Peritoneal macrophages were subjected to immunoblotting to detect the NLRP3 knockout efficiency.

**Mitigation of MSU-induced peritonitis.** C57BL/6 mice (6–8 weeks, $n = 5$ per group) were first injected with $CLAN_{mCas9/gNLRP3}$ at the dose of 1 or 4 mg kg$^{-1}$ total RNA for three injections. Mice injected with PBS, $Free_{mCas9/gNLRP3}$ (4 mg kg$^{-1}$), $CLAN_{mCas9/gNC}$ (4 mg kg$^{-1}$) were used as controls. Two days after the last $CLAN_{mCas9/gNLRP3}$ injection, all mice were intraperitoneally injected with MSU (40 mg kg$^{-1}$) to induce peritonitis. The mice were then sacrificed at 6 h after MSU injection, and the peritoneal cavities were washed with 10 ml of cold PBS. Neutrophils in the peritoneal lavage fluid were labeled with anti-CD11b and anti-Ly6G antibodies and were analyzed by FACS to detect neutrophil recruitment. The concentration of IL-1β in peritoneal lavage fluid was measured by ELISA. Peritoneal macrophages were subjected to immunoblotting to detect the NLRP3 knockout efficiency.

**Amelioration of HFD-induced T2D.** The HFD-induced T2D mouse model was established by feeding C57BL/6 mice (6 weeks) on HFD diet (60% kcal fat, D12492, Research Diets, NJ, USA) for 12 weeks and randomly allocated into 6–8 per group. Next, T2D mice were injected daily with $CLAN_{mCas9/gNLRP3}$ daily at the dose of 0.5, 1, or 2 mg kg$^{-1}$ total RNA for 3 injections and were further injected with $CLAN_{mCas9/gNLRP3}$ twice a week for 7 injections. Two days post last injection (day 30), GTT or ITT was performed to detect the therapeutic efficacy. T2D mice injected with PBS, $Free_{mCas9/gNLRP3}$ (2 mg kg$^{-1}$), or $CLAN_{mCas9/gNC}$ (2 mg kg$^{-1}$) were used as negative controls. T2D mice intraperitoneally injected daily with glyburide (500 mg kg$^{-1}$) for 30 injections were used as positive controls. Healthy control mice were fed on NCD diet (10% kcal fat, D12450, Research Diets, NJ, USA). To detect the fasting glucose, mice were fasted for 8 h. For GTT, mice were injected intraperitoneally with glucose at a dose of 1.5 g kg$^{-1}$ after fasting for 8 h. For ITT, mice were injected with 1.5 IU kg$^{-1}$ of recombinant human insulin (Gibco, Eggenstein, Germany) after fasting for 6 h. The blood samples were then obtained at different time points (0, 15, 30, 60, 120 min) for glucose measurements using a Glucosemeter (Roche, Basel, Switzerland). WAT from the epididymal fat pad was collected and washed in cold PBS. One portion of WAT was subjected to immunoblotting to analyze IL-1β or caspase p10 or NLRP3. Another portion of WAT was cultured in 12-well plates in opti-MEM supplemented with penicillin/streptomycin (Gibco, Eggenstein, Germany) for 24 h. Next, the supernatants were collected to measure IL-1β, IL-18, TNF-α, or MCP-1 by ELISA.

**Sample size and randomization.** Simple size was chosen according to the results of the preliminary experiment and animals were randomly allocated to experimental groups without blinding to investigators.

**Statistical analyses.** All values were expressed as the means ± SEM. Statistical analysis was performed using unpaired $t$-test for two groups, two-way ANOVA for multiple groups or log-rank test for survival distributions (GraphPad Software) with all the data points showing a normal distribution. $P < 0.05$ was considered significant.

## Data availability

The data that support this study are available within the article and its Supplementary Information files or available from the authors upon request.

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

## Acknowledgements

We thank R. Zhou for advices and discussions, Z. Lian for technical support. This work was supported by the National Key R&D Program of China (2017YFA0205600), the Program for Guangdong Introducing Innovative and Enterpreneurial Teams (2017ZT07S054), the National Basic Research Program of China (2015CB932100), the National Natural Science Foundation of China (51390482, 51633008 and 51728301), and the China Postdoctoral Science Foundation (2018M630953).

## Author contributions

C.X., Z.L., Y.L.L., Y.L., S.S., J.L., K.C. and Z.C. performed the experiments of this work; Z. C. and H.L. synthesized lipids and polymers. C.X., Z.L., and J.W. designed the research. C.X. and Z.L. wrote the manuscript. X.Y. and Z.G. revised the manuscript. J.W. supervised the project.

## Additional information

**Competing interests:** The authors declare no competing interests.

