## [Peer Review File · Nature Communications]

Editorial Note: This manuscript has been previously reviewed at another journal that is not operating a transparent peer review scheme. This document only contains reviewer comments and rebuttal letters for versions considered at Nature Communications. Mentions of prior referee reports have been redacted.

Reviewer #1:

None

Reviewer #2:

Remarks to the Author:

My comments have been addressed

Reviewer #3:

Remarks to the Author:

The authors have largely addressed my concerns from the first review, have added more data, and have improved the manuscript. I would suggest:

1. Including Figure R1 in the manuscript or supplementary data and noting it in the text.
2. In the discussion, adding comment about the potential side-effects of permanent NLRP3 inhibition.

Referee 2: Comments have been addressed.

Response: Thanks a lot.

Referee 3,

Comment 1: The authors have largely addressed my concerns from the first review, have added more data, and have improved the manuscript. I would suggest: Including Figure R1 in the manuscript or supplementary data and noting it in the text.

Response: Thanks for the suggestion. We have added Figure R1 into the Supplementary Information as Supplementary Fig. 14. The text has been added into the Manuscript (page 19, line 403-407).

Comment 2: The authors have largely addressed my concerns from the first review, have added more data, and have improved the manuscript. I would suggest: Including Figure R1 in the manuscript or supplementary data and noting it in the text.

Response: We have commented the potential side-effects of permanent NLRP3 inhibition in the Manuscript (page 20, line 408-409).